# Comparison of Cognitive Deterioration Between Propofol and Remimazolam Anesthesia in ApoE4 Knock-In Mouse Model

**DOI:** 10.3390/ijms26125718

**Published:** 2025-06-14

**Authors:** Jong-Ho Kim, Songyi Park, Harry Jung, Eun-Hae Lee, Eun-Seo Lee, Jae-Jun Lee, Jong-Hee Sohn

**Affiliations:** 1Department of Anesthesiology and Pain Medicine, Chuncheon Sacred Heart Hospital, College of Medicine, Hallym University, Chuncheon 24253, Republic of Korea; poik99@hallym.or.kr; 2Institute of New Frontier Research, College of Medicine, Hallym University, Chuncheon 24252, Republic of Korea; songyip3697@gmail.com (S.P.); harry_88@naver.com (H.J.); eunhaetop@naver.com (E.-H.L.); dmstj5693@gmail.com (E.-S.L.); 3Department of Neurology, Chuncheon Sacred Heart Hospital, College of Medicine, Hallym University, Chuncheon 24253, Republic of Korea

**Keywords:** propofol, remimazolam, anesthesia, general, cognitive impairments

## Abstract

Perioperative neurocognitive disorder (PND) is a concern following anesthesia, particularly in individuals at risk for Alzheimer’s disease (AD). This study compared the cognitive and pathological effects of propofol and remimazolam in a mouse model with AD following surgery. Five-month-old male ApoE4-KI mice underwent abdominal surgery under either propofol (170 mg/kg) or remimazolam (85 mg/kg) anesthesia. Cognitive function was assessed using the Morris water maze and Y-maze, and neuronal apoptosis and amyloid-beta (Aβ) deposition in the CA3 and dentate gyrus (DG) of the hippocampus were evaluated preoperatively and at 2, 4, and 7 days postoperatively. Both groups showed similar postoperative cognitive functions, with increased relative escape latency at day 2 and decreased relative spontaneous alternation at days 4 and 7. However, the neuropathological analysis revealed that propofol-induced significantly more neuronal death in the CA3 (days 4 and 7) and DG (days 2, 4, and 7), and greater Aβ accumulation in the CA3 (days 2 and 4) and DG (days 2 and 7) compared to remimazolam (*p* < 0.05). Propofol was associated with more pronounced neuropathologic changes in the hippocampus compared to remimazolam. These findings suggest remimazolam may be a safer anesthetic for patients at risk for neurodegenerative disorders, as it is associated with less severe hippocampal pathology, which is characteristic of AD.

## 1. Introduction

Perioperative neurocognitive disorder (PND) is a significant and common complication following anesthesia and surgery, particularly in older adult patients [1,2,3] and those at risk for Alzheimer’s disease (AD) or another neurodegenerative disease [4]. PND encompasses a spectrum of cognitive impairments, including memory loss, attention deficits, executive dysfunction, and personality changes that can last from days to months, and in some cases, years [5,6]. The incidence reports show that up to 25% of older adult patients develop PND one week after surgery, and approximately 10% may still have cognitive deficits three months after surgery [2].

The mechanisms underlying PND are multifactorial and not fully understood. Hypotheses include neuroinflammation, blood-brain barrier disruption, oxidative stress, and direct neurotoxicity of anesthetic agents [7,8]. Especially, anesthetic agents may exacerbate neurodegenerative processes, particularly in individuals with genetic risk factors for AD [9,10]. Preclinical studies indicate that anesthetics interact with several factors involved in AD pathogenesis at multiple levels [11].

The apolipoprotein E (ApoE) gene exists in three major isoforms—E2, E3, and E4—with the ApoE4 allele is known to be a major genetic risk factor for sporadic, late-onset AD [12,13]. Individuals carrying the ApoE4 allele have an increased risk of developing AD and tend to exhibit earlier disease onset and more severe pathology compared to non-carriers [14]. They are also more susceptible to neurocognitive decline following anesthesia and surgery [15]. ApoE4 is thought to influence amyloid-beta (Aβ) aggregation and clearance, tau phosphorylation, and neuroinflammation, thereby modulating the vulnerability of the brain to neurodegenerative processes [16]. Aβ, a cleavage product of the amyloid precursor protein (APP), accumulates abnormally in the brain in AD and is associated with synaptic dysfunction and neuronal toxicity [17]. Aβ peptides aggregate to form oligomers and plaques that disrupt synaptic function, induce oxidative stress and trigger neuroinflammatory responses, ultimately leading to neuronal death and cognitive impairment [18].

To investigate the genetic and molecular mechanisms of AD in vivo, ApoE4 knock-in (ApoE4-KI) mice were developed by replacing the endogenous murine ApoE gene with the human ApoE4 allele [19]. ApoE4-KI mice progressively develop hallmark AD-related neuropathological changes, including Aβ deposition, tau phosphorylation, synaptic loss, and gliosis, even in the absence of overt behavioral deficits in the early stages [20]. Thus, they provide valuable insights into how anesthetic agents may accelerate cognitive decline in genetically predisposed individuals [21,22].

Propofol is a widely used intravenous anesthetic agent known for its rapid onset and short duration of action. It primarily acts as a positive allosteric modulator of the gamma-aminobutyric acid type A (GABA_A_) receptor, enhancing inhibitory neurotransmission and inducing sedation, hypnosis, and amnesia [23]. Despite its favorable pharmacokinetics, propofol has been associated with several adverse effects, including hypotension, respiratory depression, and dose-dependent neurotoxicity [24,25]. Preclinical studies have shown that propofol may exacerbate Aβ aggregation, induce tau hyperphosphorylation, and trigger neuronal apoptosis, particularly in models predisposed to AD pathology [21,26,27,28].

Previous results have shown that propofol induces persistent cognitive deficits and hippocampal pathology in pre-symptomatic AD mice [21]. These effects were validated through behavioral tests, notably the Morris Water Maze (MWM) and Y-Maze, which are widely used in preclinical models to assess hippocampal function. The MWM evaluates spatial memory dependent on hippocampal integrity [29], while the Y-Maze measures short-term memory and exploratory behavior [30]. Given that these cognitive domains are frequently impaired in PND [31], both tasks are commonly employed to investigate anesthesia-related cognitive dysfunction.

Remimazolam, a newer ultra-short-acting benzodiazepine, has gained attention as a promising alternative to propofol due to its rapid metabolism and lower accumulation in the body [32]. It is metabolized by tissue esterases to an inactive metabolite, resulting in a predictable and short duration of action [33]. This benefit is attributed to its pharmacokinetic profile which, as demonstrated in studies using the HVLT-R test [34], suggests a potential for reduced cognitive dysfunction. While these findings are encouraging, there is limited research on the specific effects of remimazolam on cognitive function in populations at risk for neurodegenerative diseases, such as those with the ApoE4 allele. However, recent studies suggest it may induce cognitive dysfunction via glutamate excitotoxicity and increase Aβ levels [35]. Despite these findings, no study has directly compared the extent of Aβ accumulation between remimazolam and propofol, leaving their relative neurotoxicity unclear.

In light of these considerations, this study aimed to further investigate the cognitive effects of remimazolam compared to propofol, focusing on vulnerable populations such as those with preclinical AD. Using the ApoE4-KI mouse model, which mimics sporadic AD in humans, we evaluated the cognitive and neuropathological changes following surgery under remimazolam or propofol-induced anesthesia. Additionally, we assessed behavioral changes using established cognitive tests and examined neuropathological markers such as Aβ accumulation. This research could provide critical insights into safer anesthetic strategies to improve postoperative cognitive outcomes in at-risk patients.

To ensure a valid comparison of cognitive and neuropathological outcomes, anesthetic doses for propofol [36,37] and remimazolam [38,39,40] were selected based on published median effective dose (ED50) and 95% effective dose (ED95) values in mice. These doses were designed to achieve comparable anesthetic depth and duration, minimizing the likelihood that outcome differences resulted from unequal exposure or anesthetic potency. Further details and rationale for dose selection are provided in the Methods section.

We hypothesized that remimazolam, due to its favorable pharmacokinetic profile, would induce less postoperative cognitive impairment and attenuate AD-related neuropathological changes compared to propofol in ApoE4-KI mice.

## 2. Results

### 2.1. Body Weight

In this study, the animals were weighed preoperatively and at 2, 4, and 7 days postoperatively. Two-way analysis of variance (ANOVA) revealed a significant effect of time on weight (F = 6.508, *p* < 0.001); however, there were no significant group (F = 0.067, *p* = 0.796) or interaction effects (F = 0.286, *p* = 0.835). Postoperative weight was significantly lower than preoperative weight (preoperative: 31.05 ± 2.08 g; 2 days: 29.00 ± 2.33 g, *p* = 0.001; 4 days: 29.01 ± 2.63 g, *p* = 0.005; 7 days: 29.11 ± 2.63 g, *p* = 0.008; Tukey HSD post hoc). Weight changes over time are summarized in Table 1.

### 2.2. Behavioral Testing Results

#### 2.2.1. Morris Water Maze Test Results

Spatial learning and memory were assessed using the Morris Water Maze (MWM) test. The change in mean MWM escape latency from before and after surgery is summarized in Table 2.

Two-way ANOVA revealed no significant effect of time (F = 2.498, *p* = 0.065), group (F = 0.044, *p* = 0.833), or interaction (F = 0.025, *p* = 0.995) on absolute escape latency. For relative escape latency, there was a significant effect of time (F = 2.767, *p* = 0.046), but no significant group (F = 0.002, *p* = 0.966) or interaction effects (F = 0.120, *p* = 0.948 The post hoc test showed a significant increase in relative escape latency at 2 days postoperatively (preoperative: 1.00 ± 0.00; 2 days: 1.87 ± 1.51, Tukey HSD *p* = 0.024); although no significant differences were found at 4 days (1.47 ± 1.03, Tukey HSD *p* = 0.336) or 7 days (1.50 ± 0.81, Tukey HSD *p* = 0.357). There were no significant differences in absolute and relative escape latencies between the two groups at any postoperative time point (2, 4, and 7 days after surgery, *p* > 0.05). During the training phase, there was no significant difference in escape latency between the two groups (Figure 1).

Two-way ANOVA revealed no significant effects of time, group, or interaction on swimming speed (time: F = 0.911, *p* = 0.438; group: F = 0.001, *p* = 0.978; interaction: F = 0.306, *p* = 0.821), time spent in the four quadrants (time: F = 1.498, *p* = 0.220; group: F = 0.049, *p* = 0.825; interaction: F = 1.224, *p* = 0.305), or swim path length (time: F = 1.424, *p* = 0.240; group: F = 1.195, *p* = 0.277; interaction: F = 1.420, *p* = 0.241).

#### 2.2.2. Y-Maze Test Results

Y-maze tests were conducted to measure spatial and short-term memory. Two-way ANOVA revealed no significant effects of time (F = 1.862, *p* = 0.139), group (F = 0.049, *p* = 0.826), or interaction (F = 0.402, *p* = 0.752) on absolute spontaneous alternation. However, there was a significant effect of time on relative spontaneous alternation (F = 4.390, *p* = 0.006), with no significant group (F = 0.268, *p* = 0.606) or interaction effects (F = 0.084, *p* = 0.969). Post hoc tests revealed significant increases in relative spontaneous alternation at 4 days (1.24 ± 0.30, Tukey HSD *p* = 0.021) and 7 days postoperatively (1.28 ± 0.39, Tukey HSD *p* = 0.007); although no significant differences were found at 2 days (1.19 ± 0.34, Tukey HSD *p* = 0.092). The changes in the mean spontaneous alternation as measured by the Y-maze test before and after surgery are summarized in Table 3.

Arm-entry frequency also showed a significant effect of time (F = 23.576, *p* < 0.001); however, there were no significant groups (F = 1.432, *p* = 0.234) or interaction effects (F = 0.019, *p* = 0.996). Post hoc analysis revealed significant differences between the preoperative and postoperative days 2, 4, and 7 data (preoperative: 25.36 ± 8.28; 2 days: 12.71 ± 7.32, *p* < 0.001; 4 days: 10.59 ± 4.84, *p* < 0.001; 7 days: 14.48 ± 6.60, *p* < 0.001; Tukey HSD post hoc).

### 2.3. Histopathology

#### 2.3.1. Apoptosis

Brain tissue was collected for Terminal deoxynucleotidyl transferase dUTP nick end labeling (TUNEL) staining to measure neuronal apoptosis in the CA3 and dentate gyrus (DG) of the hippocampus (Figure 2A) at the following time points: 2 days before surgery and 2, 4, and 7 days postoperatively (after completion of behavioral tests).

Two-way ANOVA revealed significant time (F = 601.066, *p* < 0.001), group (F = 489.053, *p* < 0.001), and interaction effects (F = 330.255, *p* < 0.001) on TUNEL positivity in the CA3 region. Post hoc tests revealed significant differences between preoperative data and data obtained on postoperative days 2, 4, and 7 (preoperative: 120.35 ± 54.66; 2 days: 650.00 ± 143.92, *p* < 0.001; 4 days: 2437.50 ± 1637.45, *p* < 0.001; 7 days: 818.75 ± 387.25, *p* < 0.001; Tukey HSD post hoc). Similarly, in the DG region, two-way ANOVA showed significant effects of time (F = 752.502, *p* < 0.001), group (F = 2250.526, *p* < 0.001), and interaction (F = 560.893, *p* < 0.001). Post hoc tests also revealed significant differences between preoperative data and those obtained on postoperative days 2, 4, and 7 (preoperative: 102.13 ± 16.08; 2 days: 1687.50 ± 1143.85, *p* < 0.001; 4 days: 2150.00 ± 1859.15, *p* < 0.001; 7 days: 893.75 ± 347.89, *p* < 0.001; Tukey HSD post hoc). A summary of TUNEL positivity changes before and after surgery is displayed in Table 4.

Preoperatively, there was no significant difference between the groups. Postoperatively, neuronal apoptosis was significantly higher in the propofol group compared to that in the remimazolam group in the CA3 region on day 4 (Figure 2B; *p* < 0.001), and in the DG region on both days 2 and 4 (Figure 2C; *p* < 0.001). By postoperative day 7, TUNEL positivity decreased in both groups, although the propofol group still showed higher levels than the remimazolam group, with significant differences remaining in both the CA3 and DG regions (*p* < 0.001).

#### 2.3.2. Aβ Deposition

Amyloid burden (percentage of the area bound by Aβ 1-42 antibodies) was calculated by measuring Aβ levels in the CA3 and DG regions of the hippocampus. Image analysis was used to quantify the area of immunopositive Aβ deposition (Figure 3A).

Two-way ANOVA revealed significant time (F = 162.725, *p* < 0.001), group (F = 77.419, *p* < 0.001), and interaction (F = 27.883, *p* < 0.001) effects on Aβ levels in the CA3 region. Post-hoc tests showed significant differences between preoperative and postoperative data on days 2 and 4 (preoperative: 100.00 ± 17.25; 2 days: 207.57 ± 33.34, *p* < 0.001; 4 days: 273.30 ± 89.67, *p* < 0.001; Tukey HSD post hoc), but not on day 7 (91.95 ± 23.30, Tukey HSD *p* = 0.841). There was also significant time (F = 193.577, *p* < 0.001), group (F = 229.794, *p* < 0.001), and interaction (F = 90.544, *p* < 0.001) effects in the DG region. Post-hoc tests showed no significant differences between preoperative and postoperative day 2 day (preoperative: 100.00 ± 3.31; 2 days: 100.88 ± 19.64, Tukey HSD *p* = 0.942), but significant differences were found at day 4 (119.73 ± 3.54, Tukey HSD *p* < 0.001) and day 7 (82.23 ± 17.85, Tukey HSD *p* < 0.001). Table 5 summarizes the changes in Aβ expression levels before and after surgery.

Preoperatively, there was no significant difference between the groups. Postoperatively, Aβ levels were significantly higher in the propofol group compared to the remimazolam group on days 2 and 4 in the CA3 region (Figure 3B; *p* = 0.004, *p* < 0.001, respectively) and on days 2 in the DG region (Figure 3C; *p* < 0.001). By day 7, Aβ levels decreased in both groups; however, it remained significantly higher in the DG region in the propofol group (*p* < 0.001), with no significant difference in the CA3 region (*p* = 0.212).

## 3. Discussion

In this study, we investigated the cognitive effects of two anesthetic agents, remimazolam and propofol, in ApoE4-KI mice after surgery. In the neuropathologic findings, the propofol group showed a significant increase in Aβ deposition and neuronal apoptosis, particularly in the CA3 and DG regions of the hippocampus compared with that in the remimazolam group. Our results indicate significant differences in postoperative cognitive neuropathological outcomes between the two anesthetics, with implications for patients at risk of AD or related cognitive impairments.

Our results were consistent with previous research on PND in older adults, particularly those predisposed to neurodegenerative diseases such as patients with AD [10,41,42]. In the behavioral tests, differences were observed pre- and post-operatively; however, no significant differences were found between the remimazolam and propofol groups of ApoE4-KI mice. However, histopathologic analysis revealed significant differences between the two groups. The remimazolam group demonstrated less severe neuropathological changes, indicated by lower levels of hippocampal Aβ accumulation and neuronal apoptosis compared to that in the propofol group. These results may be attributed to the pharmacological profile of remimazolam, a benzodiazepine known for its shorter half-life and lower likelihood of inducing neurotoxicity compared to other anesthetics [32,33,34].

These changes are critical because Aβ accumulation and neuronal death are hallmarks of AD pathology that can accelerate cognitive decline [43]. Previous in vitro experiments indicate that several anesthetics can modulate Aβ peptide oligomerization and deposition, promoting the formation of amyloid plaques, a critical step in AD pathogenesis [44,45]. Similarly, our histopathologic findings suggest that propofol may exacerbate Aβ deposition and neuronal apoptosis in the hippocampus of ApoE4-KI mice, potentially reflecting the neurotoxic effects observed with isoflurane. For example, studies have shown that clinically relevant concentrations of isoflurane induce apoptosis in both human neuroglioma cell lines and murine models, an effect amplified by increased amyloid precursor protein C-terminal fragments [46,47]. This suggests that isoflurane may trigger a self-perpetuating cycle of apoptosis, Aβ peptide production and aggregation, and further apoptosis [48]. Considering the natural presence and age-related increase of Aβ peptide in the central nervous system, previous literature has suggested that certain general anesthetics administered to older adult patients may exacerbate Aβ peptide oligomerization and deposition, thereby increasing the risk of postoperative cognitive impairment [10].

Other animal studies, including those using non-AD models, have shown that anesthesia-induced cognitive decline is not solely a result of the anesthetic agent; however, it is also influenced by factors such as surgical stress and neuroinflammatory responses [49,50]. For example, Terrando et al. demonstrated that surgery combined with general anesthesia induced neuroinflammation and cognitive decline in wild-type mice, suggesting that surgery itself, along with anesthesia, contributes to cognitive impairment [7].

Previous studies using other transgenic mouse models of AD, such as 3xTg-AD [51] and APP/PS1 mice [52], have shown that general anesthetics, particularly inhalational agents such as isoflurane and sevoflurane, exacerbate cognitive impairment and promote Aβ deposition and tau phosphorylation. These studies suggest that inhalational anesthetics may trigger or accelerate the pathological processes underlying AD. For example, Xie et al. demonstrated that inhaled isoflurane significantly increased Aβ levels and induced neuronal apoptosis in the hippocampus of AD transgenic mice, contributing to cognitive decline [47]. Similarly, other studies have reported that exposure to inhaled anesthetics increases tau phosphorylation and synaptic dysfunction, key features of AD pathology [53].

In contrast, studies of propofol, a commonly used intravenous anesthetic, have reported mixed results regarding its neurocognitive effects, particularly in AD-prone models. While some research suggests that propofol is safer than inhaled anesthetics, other findings highlight its potential to induce cognitive impairment and hippocampal pathology. For example, in aged APP/PS1 transgenic mice, repeated propofol exposure did not exacerbate Aβ plaque deposition or synaptic loss [54]. However, propofol has been shown to increase Aβ accumulation and neuronal apoptosis in the hippocampus of ApoE4-KI mice [21]. Similar effects were observed in 3xTg-AD mice, where propofol anesthesia exacerbated tau hyperphosphorylation and caused persistent spatial memory deficits [55]. These findings suggest that while propofol may have a relatively safer profile than inhaled anesthetics, it still poses significant neurocognitive risks in vulnerable populations and experimental models.

Remimazolam, a newer ultrashort-acting benzodiazepine, has not been extensively studied in the context of cognitive impairment and AD pathology, and its cognitive effects vary among studies. High doses have been reported to induce cognitive dysfunction through mechanisms such as glutamate excitotoxicity, neuronal loss, and Aβ plaque formation in mice [35]. Conversely, remimazolam was found to delay memory decline in aged mice by reducing tau phosphorylation [56]. In addition, remimazolam attenuated neuroinflammation-induced cognitive impairment through its anti-inflammatory effects [57]. Our results suggest that while remimazolam may still contribute to postoperative cognitive dysfunction, it appears to present a more favorable neurocognitive profile compared to propofol, particularly in terms of reduced Aβ deposition and neuronal apoptosis in the hippocampus.

Clinical studies investigating the impact of remimazolam on postoperative cognitive function highlight the limitation of direct comparisons with propofol. However, preliminary results suggest that remimazolam may reduce postoperative cognitive dysfunction and delirium, especially in older adult patients undergoing surgery. For example, remimazolam has demonstrated effective sedation with rapid recovery and minimal residual sedative effects, which may contribute to improved postoperative cognitive outcomes [58,59] and a lower incidence of delirium than that with propofol [60]. However, other studies have reported no significant differences in the incidence of postoperative delirium between remimazolam and propofol [61]. These findings suggest that the pharmacokinetic properties of remimazolam may either provide an advantage or have a similar effect in reducing postoperative cognitive impairment or exert a similar impact compared to propofol.

These emerging clinical findings complement our study by providing evidence that remimazolam’s favorable neurocognitive profile observed in animal models may translate into clinical benefits. However, further extensive clinical trials directly comparing remimazolam and propofol in terms of cognitive outcomes are necessary to confirm these potential advantages. Given the increasing prevalence of surgeries in the aging population and the associated risks of cognitive decline, the selection of anesthetic agents with minimal neurotoxic effects is of paramount importance.

In addition to findings in older adults or genetically predisposed individuals, early-life exposure to general anesthetics has also been associated with long-term neurodevelopmental impairments in both animal models and human epidemiological studies. Studies in neonatal rodents have demonstrated that exposure to anesthetic agents such as isoflurane, sevoflurane, and propofol during critical periods of brain development can result in widespread neuronal apoptosis, disrupted synaptogenesis, and persistent cognitive deficits [62]. Similarly, epidemiological studies in pediatric populations have reported associations between early exposure to general anesthesia and subsequent impairments in language acquisition, learning ability, and overall cognitive function [63,64]. Although our study primarily focuses on aging and preclinical AD, these findings from early-life models highlight the broader neurotoxic potential of certain anesthetic agents across the lifespan. The shared pathological features suggest that overlapping molecular mechanisms may contribute to anesthesia-induced neurotoxicity in both neurodevelopmental and neurodegenerative contexts.

In preclinical AD, histopathologic changes occur before the onset of clinical symptoms, and preclinical AD can last for years or even decades [65]. Our results support this, as we observed significant histological differences without corresponding behavioral deficits. This suggests that anesthesia and surgery may accelerate the preclinical stage of AD, possibly leading to an earlier onset of cognitive symptoms. The lack of immediate behavioral changes does not preclude future cognitive decline, as the accumulation of neuropathological changes may manifest clinically over time [66].

Our study has several limitations. First, there is a relatively short observation period, focusing on cognitive function and neuropathologic changes up to 7 days after surgery. Although significant differences between remimazolam and propofol were observed during this period, it remains uncertain whether these cognitive effects persist or evolve over a longer period. Given that PND can manifest weeks to months after surgery, this short-term assessment may not fully capture the long-term cognitive impact of these anesthetics. Future research should extend the observation period to assess whether the initial cognitive impairments resolve, persist, or worsen over time. Second, the study relied on a limited number of behavioral assessments, specifically the MWM and Y-maze tests, which access hippocampal-dependent spatial and short-term memory, respectively [29,30]. These paradigms were selected because they are well-established, sensitive, and commonly used to assess cognitive domains frequently affected by PND [31]. However, PND is a heterogeneous condition that also involves other domains such as attention, executive function, and learning [67]. The inclusion of additional behavioral tests, such as the novel object recognition test [68], which assesses recognition memory and is less reliant on spatial navigation, could have helped identify non-hippocampal cognitive deficits. Expanding the cognitive testing battery in future studies would allow for a deeper exploration of the cognitive changes associated with anesthesia and surgery. Third, we did not perform direct physiological monitoring, such as oxygen saturation, respiratory rate, or arterial blood gas analysis, during the anesthetic period, aside from maintaining body temperature at 37 °C. Although no signs of distress or delayed recovery were observed, and the depth of anesthesia was confirmed using the righting reflex and withdrawal reflex, the lack of continuous homeostatic data limits our ability to exclude the influence of systemic physiological disruptions on cognitive outcomes [69]. Fourth, although TUNEL staining was used to assess neuronal apoptosis, it was not accompanied by a nuclear counterstain such as 4′,6-diamidino-2-phenylindole (DAPI). The lack of nuclear markers limits the precise localization of DNA fragmentation within cell nuclei and distinguishes apoptotic cells from necrotic or artifactual signals [70]. While the distribution of TUNEL-positive signals supports apoptotic activity, the absence of a nuclear marker reduces the morphological resolution and interpretability of the histological findings. Future histological analyses should include nuclear or cell-specific counterstains to enhance structural validation and accuracy. Fifth, this study primarily focuses on key neuropathological markers like Aβ deposition and neuronal apoptosis but does not address other important molecular mechanisms, such as neuroinflammation and oxidative stress, which are known to play significant roles in cognitive decline after surgery [71]. The absence of data on neuroinflammatory cytokines or markers of oxidative stress limits the ability to draw firm conclusions about the underlying mechanisms driving the observed cognitive impairments. Future studies should integrate these molecular analyses to better understand the broader biological processes contributing to anesthesia-induced cognitive decline.

Our findings suggest that remimazolam may be a safer option for patients with pre-existing cognitive vulnerabilities because it induces less severe neurodegenerative changes than propofol. However, further research is needed to confirm these findings in clinical settings and to explore the long-term cognitive effects of different anesthetic agents in vulnerable populations.

## 4. Materials and Methods

### 4.1. Experimental Animals

Having received approval from the Institutional Animal Care and Use Committee of Hallym University (No. R1 2021-74), all experimental procedures in this study were performed in accordance with the committee’s guidelines. Forty-five-month-old male Apoe^tm1.1(APOE*4)Adiuj^ (ApoE4-KI) mice (JAX stock #027894), a preclinical model of AD, were used. These mice exhibit a progressive development of AD-related neuropathological changes. While not aged, mice at this age begin to exhibit early neuropathological features associated with AD, such as Aβ accumulation, synaptic impairment, gliosis, and neuronal apoptosis. Despite these changes, cognitive functions like memory and problem-solving remain within normal parameters. This stage was selected to explore the early molecular and cellular alterations associated with ApoE4 expression, consistent with previous studies investigating the early pathophysiology of AD [19,20,21,72]. The ApoE4-KI mice used in these experiments were bred from mice purchased from the Central Experimental Animal Center (Seoul, Republic of Korea). The animals were housed under a 12-h light/dark cycle (lights on at 8:00 a.m.) with ad libitum access to food and water. The housing environment was maintained at a constant temperature of 23 ± 2 °C and a humidity of 55 ± 10%.

### 4.2. Experimental Timeline

This study included a total of 8 groups (n = 5 per group), divided into two anesthetic groups: propofol (Group P) and remimazolam (Group R). The cognitive function of ApoE4-KI mice was evaluated preoperatively and at 2, 4, and 7 days postoperatively. Prior to the experiment, all mice were trained twice a day for five days to acclimatize them to the water maze test. Prior to the experiment, all mice were trained in the water maze twice a day (4 h apart) for 5 days. Two days before anesthesia and surgery, all mice were assessed using the Y-maze test (to evaluate spatial working memory and short-term memory) and, 60 min later, the MWM test (to assess spatial learning and memory). The preoperative group (Group P, n = 5/Group R, n = 5) was then sacrificed by perfusion at the end of the testing, while the remaining mice were operated under propofol or remifentanil anesthesia. On postoperative day 2, all mice underwent Y-maze testing, followed by MWM testing at 60-min intervals, and the 2-day group was sacrificed for histologic examination. The 4- and 7-day groups were further evaluated on the same schedule. The overall experimental timeline and group design are shown in Figure 4. To minimize bias, all behavioral testing, histological processing, and data analysis were performed by experimenters who were blinded to the treatment groups. Mice were assigned randomized ID codes independent of group allocation, and group information was withheld from the personnel conducting behavioral assessments and histological quantifications until after data collection was complete.

### 4.3. Anesthesia and Surgery

ApoE4-KI mice used in this experiment were anesthetized via tail vein injection with either 170 mg/kg of propofol (Anepol; Hana Pharmaceuticals, Seoul, Republic of Korea) or 85 mg/kg of remimazolam (ByFavo; Hana Pharmaceuticals, Seoul, Republic of Korea) (Figure 5B). The reported median effective dose (ED50) for intravenous (IV) administration of propofol is 25 mg/kg [36], while the 95% effective dose (ED95) of propofol is approximately twofold higher [37]. In this study, propofol was administered with an induction dose of 50 mg/kg, followed by twelve 10 mg/kg doses every 10 min to maintain general anesthesia. The ED95 for intraperitoneal (IP) administration of remimazolam is reported as 45 mg/kg [38]; however, IP injection typically results in lower bioavailability compared to IV administration [39]. In this study, remimazolam was administered with an induction dose of 25 mg/kg, approximately the dose required to achieve a rapid loss of righting reflex in mice (15–30 mg/kg) [40], followed by twelve 5 mg/kg doses every 10 min using the same protocol. Anesthetic depth was assessed by monitoring the righting reflex (the ability to return to an upright position from a supine position) and the pedal withdrawal reflex (absence of response to toe pinch), which was confirmed for 5 min after induction. The body temperature of the mice was maintained at a constant 37 °C throughout the procedure. Anesthesia was maintained steady, and the total anesthesia time did not exceed 2 h. The surgical area was shaved and disinfected with povidone-iodine prior to incision. A 1.5 cm midline abdominal incision was then made, and the viscera were gently manipulated with a sterile probe for approximately 3 min (Figure 5C). Sterile 4-0 silk sutures (Ailee Co., Ltd., Busan, Republic of Korea) were used to close the lining of the peritoneum and skin. All surgeries on mice were performed by an experienced surgeon and each surgery lasted approximately 10 min. Postoperatively, the mice recovered on heating pads to maintain body temperature and were safely returned to their original cages once they were sufficiently stabilized (Figure 5D).

### 4.4. Behavioral Testing Protocol

#### 4.4.1. Morris Water Maze Test Protocol

Mice underwent five acquisition trials daily for five consecutive days. Each trial consisted of a maximum swim time of 120 s, a 5-s stay on the submerged platform, and a 4-h inter-trial interval. The MWM test was conducted 2 days before surgery and on postoperative days 2, 4, and 7. The testing apparatus consisted of a 100 cm diameter circular pool filled with opaque water (21–22 °C) achieved by adding white paint. Mice were trained to locate a hidden platform (8 cm diameter) submerged 1 cm below the water’s surface. Visual cues were positioned on the four walls of the pool, 0.5 m from the center, to aid spatial orientation. All movements were video-tracked using the Noldus EthoVision XT system (Noldus Information Technology, Leesburg, VA, USA), and swimming speed, time spent in each quadrant, path length, and escape latency were precisely recorded. Escape latency was analyzed as both absolute and relative values, with the latter calculated as a ratio relative to the preoperative baseline.

#### 4.4.2. Y-Maze Test Protocol

Mice were placed individually in a white Y-maze, with the three arms designated A, B, and C (each arm measuring 40 cm in length, 12 cm in height, and 10 cm in width). Each mouse was placed at the end of a randomly selected arm and allowed to explore the maze freely for 5 min. Entries in all three arms and the central area were recorded using a video-tracking system. Maximum possible alternations were defined as the total number of arm entries minus two since the first two entries do not constitute a complete alternation [73]. Alternation behavior was defined as consecutive entries in all three arms. Spontaneous alternation, calculated as an alternation index (number of alternations/maximum possible alternations × 100%), was the primary outcome measure. The mean alternation percentage served as the absolute spontaneous alternation value, while the ratio of this value to the preoperative value was defined as the relative spontaneous alternation value.

### 4.5. Histological Analysis

#### 4.5.1. Hematoxylin and Eosin Stain (H&E)

Mice were sacrificed 2 days preoperatively and on postoperative days 2, 4, and 7. At each time point, mice were sacrificed immediately following the completion of behavioral testing (MWM and Y-maze), and their brains were immediately perfused. The brains were then fixed in 4% (*v*/*v*) paraformaldehyde for 24 h, cryoprotected in 30% (*w*/*v*) sucrose solution, and embedded in optimal cutting temperature (OCT) compound for cryosectioning (10 μm). Finally, H&E staining was performed to analyze hippocampal morphology and size.

#### 4.5.2. TUNEL Assay

In situ detection of DNA fragmentation was performed using the Dead End Fluorometric TUNEL assay (Promega, Madison, WI, USA). Specifically, tissue sections were permeabilized with 0.2% (*v*/*v*) Triton X-100 for 5 min at room temperature. Following washing, sections were incubated with a mixture of recombinant terminal deoxynucleotidyl transferase (rTdT) and equilibration buffer for 1 h at 37 °C in the dark (chamber covered with aluminum foil). The reaction was terminated by incubation with TdT stop buffer for 15 min, followed by washing with 1× phosphate-buffered saline (PBS). Sections were then counterstained with 4′,6-diamidino-2-phenylindole (DAPI, 1 μg/mL; Molecular Probes, Eugene, OR, USA) for 10 min and rinsed three times with 1 × PBS (5 min each). Finally, eight coronal hippocampal sections from each ApoE4-KI brain were analyzed using an Olympus fluorescence microscope (Tokyo, Japan) by an observer blinded to the experimental design, and the number of TUNEL-positive cells was quantified.

#### 4.5.3. Immunohistochemistry (IHC)

Animals in each group (n = 5) were deeply anesthetized with 2.2% (*v*/*v*) isoflurane (Hana Pharmaceuticals, Inc., Gyeonggi-do, Republic of Korea), followed by transcardiac perfusion with physiological saline and then 4% (*v*/*v*) paraformaldehyde in 0.1 M phosphate buffer (pH 7.4). Brains were post-fixed overnight at 4 °C in 4% (*v*/*v*) paraformaldehyde and subsequently embedded in OCT compound. Frozen tissue blocks were sectioned at 10 μm using a Leica cryostat microtome (Wetzlar, Germany). Endogenous peroxidase activity was quenched by incubation in 0.3% (*v*/*v*) hydrogen peroxide for 15 min, followed by blocking with 2% (*v*/*v*) horse serum for 1 h to prevent non-specific antibody binding. Sections were then incubated overnight at 4 °C with a primary antibody against Aβ 1-42 (1:100 dilution, ab10148, Abcam, Cambridge, UK), followed by washing at room temperature. Sections were subsequently incubated with horseradish peroxidase (HRP)-conjugated secondary antibody (1:250 dilution, Thermo Fisher Scientific Inc., Waltham, MA, USA) for 1 h and rewashed. To visualize HRP activity, 3,3′-Diaminobenzidine (DAB) was used as the chromogen. Tissues were washed thrice with 1 × PBS (5 min each) and counterstained with Mayer’s hematoxylin. Finally, stained hippocampal tissue was examined using a light microscope (Carl Zeiss Microscopy GmbH, Zeiss, Oberkochen, Germany), and the number of positive cells was quantified using ImageJ software (ImageJ 1.49v, National Institutes of Health, Bethesda, MD, USA).

### 4.6. Statistical Analysis

All data are presented as mean ± SD. Two-way ANOVA was used to analyze behavioral test and histological data, with a minimum of five mice per group to ensure sufficient statistical power for detecting intergroup differences. Power analysis, performed using G*Power software (version 3.1.9.4, Heinrich-Heine-Universität, Düsseldorf, Germany), confirmed that this sample size allowed for the detection of differences between groups under the following conditions: Cohen’s f(V) = 0.95, α = 0.05, and 1-β = 0.90. Specifically, the analysis indicated that a total of 32 animals (4 per group across 8 groups) would be required to achieve 90% power under these assumptions. A total of 40 mice were used for behavioral testing, with five mice per group sacrificed at each time point. Post hoc power analysis using the same parameters indicated an actual power of 0.99, further confirming that the sample size was adequate for detecting group differences. Following two-way ANOVA, the Student’s *t*-test or Tukey’s HSD post hoc test was performed for comparisons between groups at the same or different time points. Statistical significance was set at *p*-value < 0.05. All statistical analyses were performed using SPSS software (SPSS for Windows, version 23.0; SPSS, Chicago, IL, USA).

## 5. Conclusions

In conclusion, while both remimazolam and propofol contribute to postoperative neurocognitive decline, the severity of neurodegenerative changes appears to be greater with propofol. The differences in histopathologic changes suggest that remimazolam may be a safer anesthetic option for patients with cognitive impairment. Based on these findings, it is recommended that anesthetic choice be carefully considered in patients at risk for neurodegeneration. Further clinical studies are warranted to confirm these results and explore long-term cognitive outcomes in at-risk human populations.

## Figures and Tables

**Figure 1 ijms-26-05718-f001:**
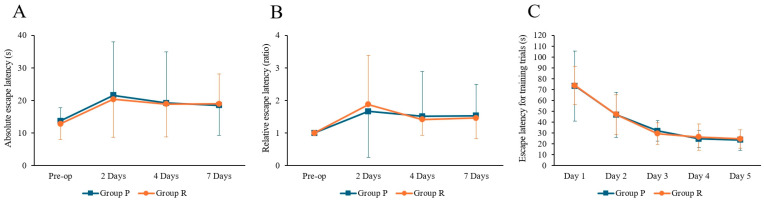
Results of Morris water maze tests in ApoE4 knock-in mice under Propofol and Remimazolam anesthesia. (**A**) No significant differences were observed between the two groups in absolute escape latency. (**B**) No significant differences were observed between the two groups in relative escape latency. (**C**) During the training trials, there were no significant differences in escape latency between the two groups. Each value represents the mean ± SD. Group P: propofol group, Group R: remimazolam group.

**Figure 2 ijms-26-05718-f002:**
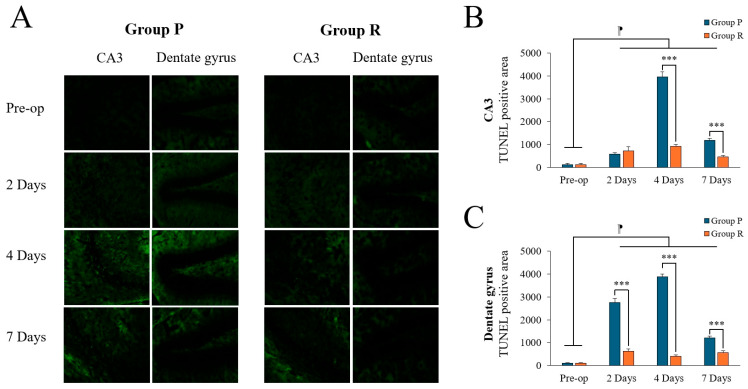
Terminal deoxynucleotidyl transferase dUTP nick end labeling (TUNEL) positivity in the hippocampal CA3 and dentate gyrus (DG) regions of ApoE4-KI mice under propofol and remimazolam anesthesia. (**A**) TUNEL staining (green) shows apoptotic cells in the CA3 and DG regions at preoperative, 2, 4, and 7 days postoperatively (scale bar = 100 µm). (**B**) Quantification of TUNEL-positive areas in the CA3 region showed no significant difference between Group P and Group R at preoperative and 2 days after surgery. However, significant differences were found at 4 and 7 days after surgery (*** *p* < 0.001). In addition, significant differences were found between preoperative and postoperative days 2, 4, and 7 (^⁋^ *p* < 0.001). (**C**) Quantification of TUNEL-positive areas in the DG region showed no significant difference at the preoperative stage. However, significant differences were observed between Group P and Group R on days 2, 4, and 7 after surgery (*** *p* < 0.001). Significant differences were also observed between preoperative and postoperative days 2, 4, and 7 (^⁋^ *p* < 0.001). Each value represents the mean ± SD. Group P: propofol group, Group R: remimazolam group.

**Figure 3 ijms-26-05718-f003:**
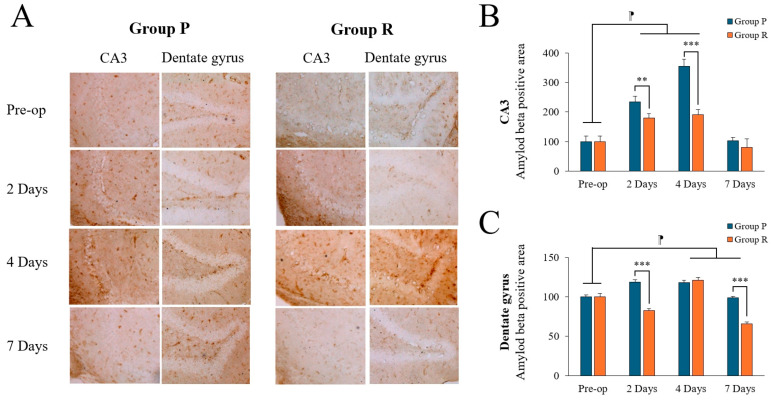
Amyloid-beta (Aβ) expression levels in the hippocampal CA3 and dentate gyrus (DG) regions of ApoE4-KI mice under propofol and remimazolam anesthesia. (**A**) Aβ deposits (brown) are visible in the CA3 and DG regions at preoperative, 2, 4, and 7 days after surgery (scale bar = 100 µm). (**B**) Quantification of Aβ deposits in the CA3 region showed no significant differences between Group P and Group R at 2 and 4 days after surgery, with higher levels in Group P. However, no significant difference was observed between the two groups at 7 days after surgery (** *p* < 0.01, *** *p* < 0.001). In addition, significant differences were observed between the preoperative and 2 and 4 days after surgery (^⁋^ *p* < 0.001). (**C**) Aβ deposition in the DG region showed no significant differences preoperatively, but significant increases were observed in Group P compared to Group R at 2 and 7 days postoperatively (*** *p* < 0.001). Significant differences were observed between the preoperative and postoperative days 4, and 7 (^⁋^ *p* < 0.001). Each value represents the mean ± SD. Group P: propofol group, Group R: remimazolam group.

**Figure 4 ijms-26-05718-f004:**
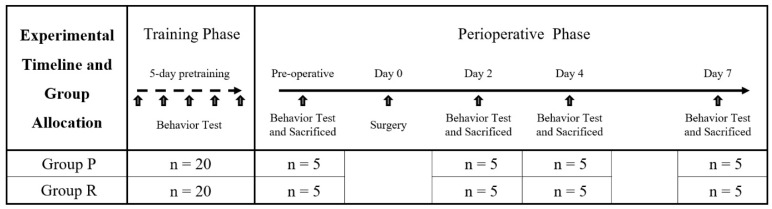
Experimental timeline and group allocation. All mice underwent 5 days of pre-training with twice-daily Morris water maze (MWM) sessions. On the preoperative day, Y-maze and MWM tests were conducted, after which mice were sacrificed (n = 5 per group) for histological analysis. The remaining mice underwent surgery, under either propofol (Group P) or remimazolam (Group R) anesthesia. Postoperatively, behavioral tests and sacrifice were conducted on days 2, 4, and 7 (n = 5 per group at each time point).

**Figure 5 ijms-26-05718-f005:**
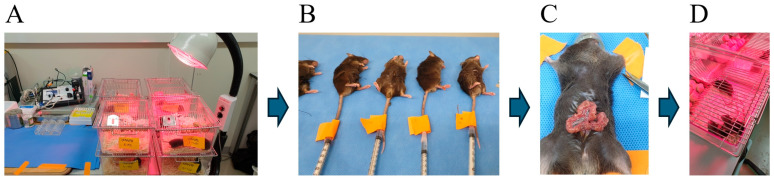
Experimental animals, anesthesia, and surgical procedures. (**A**) Animals were housed at the Hallym Clinical Translational Research Center with ad libitum access to food and water. (**B**) General anesthesia was induced with 170 mg/kg propofol or 85 mg/kg remimazolam administered by tail vein injection; the total duration of anesthesia was up to 2 h. (**C**) Surgery commenced with the creation of a median abdominal incision of approximately 1.5 cm in length into the peritoneal cavity. The viscera were then gently manipulated. (**D**) After surgery, mice were placed on heated pads and returned to their cages after recovery.

**Table 1 ijms-26-05718-t001:** Differences in body weight between the propofol and remimazolam groups.

	Group P	Group R	*p*-Value
Pre-operative	31.34 ± 1.39	30.75 ± 2.59	0.376
2 Days ^‡^	29.03 ± 1.69	28.98 ± 2.88	0.943
4 Days ^‡^	28.77 ± 1.39	29.26 ± 3.58	0.645
7 Days ^‡^	29.25 ± 1.53	28.96 ± 3.58	0.788

Each value is presented as a mean ± SD (g). Two-way ANOVA revealed a significant effect of time (F = 6.508, *p* < 0.001). ^‡^ *p* < 0.01, two-way ANOVA, and the Tukey HSD post-hoc test. Group P: propofol group, Group R: remimazolam group.

**Table 2 ijms-26-05718-t002:** Differences in mean escape latency of the Morris water maze test between the two groups before and after surgery.

	Group P	Group R	*p*-Value
A. Absolute escape latency (s)			
Pre-operative	13.68 ± 4.10	12.79 ± 4.74	0.596
2 Days	21.59 ± 16.40	20.38 ± 11.76	0.832
4 Days	19.20 ± 15.78	18.91 ± 10.17	0.957
7 Days	18.48 ± 9.18	18.97 ± 9.21	0.913
B. Relative escape latency			
Pre-operative	1.00 ± 0.00	1.00 ± 0.00	
2 Days ^†^	1.67 ± 1.42	1.87 ± 1.51	0.717
4 Days	1.51 ± 1.37	1.41 ± 0.48	0.812
7 Days	1.53 ± 0.97	1.46 ± 0.63	0.849

Each value is presented as a mean ± SD (s). A: Two-way ANOVA revealed no significant effects of time, group, or interaction on absolute escape latency. B: Two-way ANOVA revealed a significant effect of time on relative escape latency (F = 2.767, *p* = 0.046), with no significant group or interaction effects. ^†^ *p* < 0.05, two-way ANOVA, and the Tukey HSD post-hoc test. Group P: propofol group, Group R: remimazolam group.

**Table 3 ijms-26-05718-t003:** Differences in spontaneous alternation of the Y-maze test between the two groups before and after surgery.

	Group P	Group R	*p*-Value
A. Absolute spontaneous alternation (%)			
Pre-operative	48.65 ± 13.31	50.45 ± 12.68	0.716
2 Days	55.60 ± 14.74	57.75 ± 12.92	0.603
4 Days	54.87 ± 15.27	57.28 ± 8.59	0.521
7 Days	59.84 ± 19.14	55.64 ± 17.41	0.499
B. Relative spontaneous alternation			
Pre-operative	1.00 ± 0.00	1.00 ± 0.00	
2 Days	1.16 ± 0.38	1.23 ± 0.30	0.620
4 Days ^†^	1.22 ± 0.30	1.28 ± 0.31	0.664
7 Days ^†^	1.27 ± 0.37	1.28 ± 0.44	0.986

Each value is presented as a mean ± SD (%). A: Two-way ANOVA revealed no significant effects of time, group, or interaction on absolute spontaneous alternation. B: Two-way ANOVA revealed a significant effect of time on relative spontaneous alternation (F = 4.390, *p* = 0.006), with no significant group or interaction effects. ^†^ *p* < 0.05, two-way ANOVA, and the Tukey HSD post-hoc test. Group P: propofol group, Group R: remimazolam group.

**Table 4 ijms-26-05718-t004:** Differences in TUNEL positivity levels in the hippocampal region between the propofol and remimazolam groups.

	Group P	Group R	*p*-Value
A. Absolute spontaneous alternation (%)			
Pre-operative	116.75 ± 57.64	123.96 ± 60.13	0.868
2 Days ^⁋^	575.00 ± 64.55	725.00 ± 170.78	0.179
4 Days ^⁋^	3962.50 ± 217.47	912.50 ± 85.39	<0.001
7 Days ^⁋^	1175.00 ± 95.74	462.50 ± 47.87	<0.001
B. Hippocampal dentate gyrus (DG) region			
Pre-operative	102.00 ± 16.08	102.25 ± 18.57	0.984
2 Days ^⁋^	2750.00 ± 177.95	625.00 ± 104.08	<0.001
4 Days ^⁋^	3887.50 ± 110.87	412.50 ± 47.87	<0.001
7 Days ^⁋^	1212.50 ± 85.39	575.00 ± 64.55	<0.001

Each value is presented as a mean ± SD (g). A: Two-way ANOVA revealed significant effects of time and group on TUNEL positivity in the hippocampal CA3 region (time: F = 601.066, *p* < 0.001; group: F = 489.053, *p* < 0.001; interaction: F = 330.255, *p* < 0.001). Significant differences were observed between preoperative data and those obtained on postoperative days 2, 4, and 7. B: Two-way ANOVA revealed significant effects of time and group on TUNEL positivity in the DG region (time: F = 752.502, *p* < 0.001; group: F = 2250.526, *p* < 0.001; interaction: F = 560.893, *p* < 0.001). Significant differences were observed between preoperative data and postoperative days 2, 4, and 7. ^⁋^ *p* < 0.001, two-way ANOVA and the Tukey HSD post-hoc test. Group P: propofol group, Group R: remimazolam group.

**Table 5 ijms-26-05718-t005:** Differences in the hippocampal CA3 Amyloid-beta (Aβ) expression levels between propofol and remimazolam group.

	Group P	Group R	*p*-Value
A. Hippocampal CA3 region			
Pre-operative	116.75 ± 57.64	123.96 ± 60.13	0.868
2 Days ^⁋^	575.00 ± 64.55	725.00 ± 170.78	0.179
4 Days ^⁋^	3962.50 ± 217.47	912.50 ± 85.39	<0.001
7 Days	1175.00 ± 95.74	462.50 ± 47.87	<0.001
B. Hippocampal dentate gyrus (DG) region			
Pre-operative	102.00 ± 16.08	102.25 ± 18.57	0.984
2 Days	2750.00 ± 177.95	625.00 ± 104.08	<0.001
4 Days ^⁋^	3887.50 ± 110.87	412.50 ± 47.87	<0.001
7 Days ^⁋^	1212.50 ± 85.39	575.00 ± 64.55	<0.001

Each value is presented as a mean ± SD (g). A: Two-way ANOVA revealed significant effects of time and group on Aβ expression levels in the hippocampal CA3 region (time: F = 162.725, *p* < 0.001; group: F = 77.419, *p* < 0.001; interaction: F = 27.883, *p* < 0.001). Significant differences were observed between preoperative data and those obtained on postoperative days 2, and 4. B: Two-way ANOVA also revealed significant effects of time and group on Aβ expression levels in the DG region (time: F = 193.577, *p* < 0.001; group: F = 229.794, *p* < 0.001; interaction: F = 90.544, *p* < 0.001). Significant differences were observed between preoperative data and postoperative days 4 and 7. ^⁋^ *p* < 0.001, two-way ANOVA with Tukey HSD post-hoc test. Group P: propofol group, Group R: remimazolam group.

## Data Availability

The authors confirm that the data supporting the findings of this study are available within the article.

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
