# Peer review of "Comparison of Cognitive Deterioration Between Propofol and Remimazolam Anesthesia in ApoE4 Knock-In Mouse Model"

_ijms, 2025, doi:10.3390/ijms26125718_

Round 1
Reviewer 1 Report
Comments and Suggestions for Authors
In this manuscript, Kim et al describe a head to head comparison of propofol and remimazolam on PND in an ApoE KI mouse model. Overall I am highly supportive of publication of this manuscript, given the novelty of investigating remimazolam. In general the data are well presented and I have few comments on the results. I have a few comments below, which I believe should be relatively easy for the authors to address in revision.
- The authors need to clearly justify the age of their model. A 5 month old mouse does not meet criteria for aged, and while the ApoE KI makes a plausible model of AD, it is important to make this argument in a robust fashion to emphasize the significance of the work.
- The ApoE KI and the putative role of Abeta should be expanded on in the introduction so that readers who are less familiar with the biology of cognitive decline can have a more helpful background.
- It would be ideal to have a clear hypothesis statement at the end of the introduction.
- It would be helpful if the authors explained the choice of dose for both propofol and remimazolam in the introduction. Were these doses chosen on the basis of literature or pilot experiments, and if the latter I would suggest describing these pilot experiments in supplementary section, it would be very helpful to the rest of the field. How do we know that these are comparable doses in terms of sedative-hypnotic effects—said another way, how do we know that the results aren’t simply the product of overly high propofol doses relative to remimazolam given that most anesthetic toxicity phenomenon are clearly dose-responsive.
- Why did the authors choose MWM and Y-Maze as assessments of PND? It is important to make a case that these are the right tests, and a limitation is that these were the only tests conducted—which should be acknowledged.
- Did the authors do any testing for homeostatic disruptions as a result of anesthesia, typically temperature, pulse ox, respiratory rate are easy to measure in realtime. If not, some blood gases showing that physiology is not overly disarrayed at the chosen dose would greatly strengthen the conclusions of the manuscript.
- The n for behavioral testing is rather low. For negative results it is important to do and show the results of post-hoc power analysis.
- It would have been better to have TUNEL staining accompanied by DAPI or some similar cellular marker. These experiments cannot be redone, so this should be acknowledged as a limitation.
- Can the authors please describe the measures that were taken to ensure that experimenters and those who conducted the analysis were blind to experimental condition.
- It would be nice to see some reference in discussion to early life anesthesia neurotoxicity and possible relevance of these findings.
- A clear accounting of limitations of the work as the next to last paragraph in the discussion section would improve the overall quality of the wokr.
Author Response
Dear Reviewer 1,
Please find attached a revised version of our manuscript, “Comparison of cognitive deterioration between propofol and remimazolam anesthesia in ApoE4 knock-in mouse model”.
We thank you for your thoughtful suggestions regarding the original version of our paper; most of the suggested changes have been incorporated into the revision.
All revisions are described in detail in the order mentioned in the review, following your comments. We believe that the revisions have greatly improved the manuscript and hereby submit the revised version for publication.
Please note that the revised sections in the manuscript have been highlighted in yellow.
Thank you for your time and consideration.
In this manuscript, Kim et al describe a head to head comparison of propofol and remimazolam on PND in an ApoE KI mouse model. Overall I am highly supportive of publication of this manuscript, given the novelty of investigating remimazolam. In general the data are well presented and I have few comments on the results. I have a few comments below, which I believe should be relatively easy for the authors to address in revision.
- The authors need to clearly justify the age of their model. A 5 month old mouse does not meet criteria for aged, and while the ApoE KI makes a plausible model of AD, it is important to make this argument in a robust fashion to emphasize the significance of the work.
We thank the reviewer for their comments and suggestions, which have helped us improve our manuscript.
As pointed out, the justification for the use of five-month-old ApoE4-KI mice as a model of Alzheimer’s disease (AD) required further clarification. While five-month-old mice do not qualify as aged animals, we selected this time point based on previous studies indicating that ApoE4-KI mice begin to exhibit early AD-like neuropathological changes—such as synaptic deficits, gliosis, and tau phosphorylation—at this age, even though overt cognitive decline is not yet apparent. This enables us to investigate early, preclinical alterations associated with the ApoE4 genotype before the onset of behavioral symptoms, thereby providing insight into early-stage mechanisms of AD pathogenesis.
Following the reviewer's recommendation, we have revised the text in the Materials and Methods as follows:
“Forty five-month-old male Apoetm1.1(APOE*4)Adiuj (ApoE4-KI) mice (JAX stock #027894), a preclinical model of AD, were used. These mice exhibit a progressive development of AD-related neuropathological changes. While not aged, mice at this age begin to exhibit early neuropathological features associated with AD, such as Aβ accumulation, synaptic impairment, gliosis, and neuronal apoptosis. Despite these changes, cognitive functions like memory and problem-solving remain within normal parameters. This stage was selected to explore the early molecular and cellular alterations associated with ApoE4 expression, consistent with previous studies investigating the early pathophysiology of AD [19-21,72].”
(page 13, lines 458 – 466)
- The ApoE KI and the putative role of Abeta should be expanded on in the introduction so that readers who are less familiar with the biology of cognitive decline can have a more helpful background.
We thank the reviewer for their comments and suggestions, which have helped us improve our manuscript.
As pointed out, the introduction required additional background on the ApoE knock-in (KI) model and the role of amyloid-beta (Aβ) in cognitive decline to assist readers less familiar with the biology underlying neurodegeneration and Alzheimer’s disease.
Following the reviewer's recommendation, we have revised the text in the Introduction as follows:
“The apolipoprotein E (ApoE) gene exists in three major isoforms—E2, E3, and E4—with the ApoE4 allele is known to be a major genetic risk factor for sporadic, late-onset AD [12,13]. Individuals carrying the ApoE4 allele have an increased risk of developing AD and tend to exhibit earlier disease onset and more severe pathology compared to non-carriers [14]. They are also more susceptible to neurocognitive decline following anesthesia and surgery [15]. ApoE4 is thought to influence amyloid-beta (Aβ) aggregation and clearance, tau phosphorylation, and neuroinflammation, thereby modulating the vulnerability of the brain to neurodegenerative processes [16]. Aβ, a cleavage product of the amyloid precursor protein (APP), accumulates abnormally in the brain in AD and is associated with synaptic dysfunction and neuronal toxicity [17]. Aβ peptides aggregate to form oligomers and plaques that disrupt synaptic function, induce oxidative stress, and trigger neuroinflammatory responses, ultimately leading to neuronal death and cognitive impairment [18].
To investigate the genetic and molecular mechanisms of AD in vivo, ApoE4 knock-in (ApoE4-KI) mice were developed by replacing the endogenous murine ApoE gene with the human ApoE4 allele [19]. ApoE4-KI mice progressively develop hallmark AD-related neuropathological changes, including Aβ deposition, tau phosphorylation, synaptic loss, and gliosis, even in the absence of overt behavioral deficits in the early stages [20]. Thus, they provide valuable insights into how anesthetic agents may accelerate cognitive decline in genetically predisposed individuals [21,22].”
(page 2, lines 52 – 71)
- It would be ideal to have a clear hypothesis statement at the end of the introduction.
We thank the reviewer for their comments and suggestions, which have helped us improve our manuscript.
As pointed out, the original introduction lacked a clear hypothesis statement, which is important for guiding the reader and framing the study’s objectives.
Following the reviewer’s recommendation, we have revised the final paragraph of the Introduction as follows:
“We hypothesized that remimazolam, due to its favorable pharmacokinetic profile, would induce less postoperative cognitive impairment and attenuate AD-related neuropathological changes compared to propofol in ApoE4-KI mice.”
(page 3, lines 115 – 117)
- It would be helpful if the authors explained the choice of dose for both propofol and remimazolam in the introduction. Were these doses chosen on the basis of literature or pilot experiments, and if the latter I would suggest describing these pilot experiments in supplementary section, it would be very helpful to the rest of the field. How do we know that these are comparable doses in terms of sedative-hypnotic effects—said another way, how do we know that the results aren’t simply the product of overly high propofol doses relative to remimazolam given that most anesthetic toxicity phenomenon are clearly dose-responsive.
We thank the reviewer for their comments and suggestions, which have helped us improve our manuscript.
As pointed out, the rationale for selecting the specific doses of propofol and remimazolam should be explained in the Introduction to provide clearer context for readers. Although detailed dosing protocols are described in the Methods section, we agree that briefly introducing the basis for dose selection in the Introduction enhances clarity and scientific transparency.
Following the reviewer’s recommendation, we have revised the text in the Introduction as follows:
“To ensure a valid comparison of cognitive and neuropathological outcomes, anesthetic doses for propofol [36,37] and remimazolam [38-40] were selected based on published median effective dose (ED50) and 95% effective dose (ED95) values in mice. These doses were designed to achieve comparable anesthetic depth and duration, minimizing the likelihood that outcome differences resulted from unequal exposure or anesthetic potency. Further details and rationale for dose selection are provided in the Methods section.”
(page 3, lines 109 – 114)
- Why did the authors choose MWM and Y-Maze as assessments of PND? It is important to make a case that these are the right tests, and a limitation is that these were the only tests conducted—which should be acknowledged.
We thank the reviewer for their comments and suggestions, which have helped us improve our manuscript.
As pointed out, it is important to justify the choice of behavioral tests used to assess PND and acknowledge the limitation of relying solely on MWM and Y-Maze.
Following the reviewer’s recommendation, we have revised the Introduction and Discussion sections as follows:
“Previous results have shown that propofol induces persistent cognitive deficits and hippocampal pathology in pre-symptomatic AD mice [21]. These effects were validated through behavioral tests, notably the Morris Water Maze (MWM) and Y-Maze, which are widely used in preclinical models to assess hippocampal function. The MWM evaluates spatial memory dependent on hippocampal integrity [29], while the Y-Maze measures short-term memory and exploratory behavior [30]. Given that these cognitive domains are frequently impaired in PND [31], both tasks are commonly employed to investigate anesthesia-related cognitive dysfunction.”
(page 2, lines 81 – 88)
“Second, the study relied on a limited number of behavioral assessments, specifically the MWM and Y-maze tests, which access hippocampal-dependent spatial and short-term memory, respectively [29,30]. These paradigms were selected because they are well-established, sensitive, and commonly used to assess cognitive domains frequently affected in PND [31]. However, PND is a heterogeneous condition that also involves other domains such as attention, executive function, and learning [67]. The inclusion of additional behavioral tests, such as the novel object recognition test [68], which assesses recognition memory and is less reliant on spatial navigation, could have helped identify non-hippocampal cognitive deficits. Expanding the cognitive testing battery in future studies would allow for a deeper exploration of the cognitive changes associated with anesthesia and surgery.”
(page 12, lines 415 – 426)
- Did the authors do any testing for homeostatic disruptions as a result of anesthesia, typically temperature, pulse ox, respiratory rate are easy to measure in realtime. If not, some blood gases showing that physiology is not overly disarrayed at the chosen dose would greatly strengthen the conclusions of the manuscript.
We thank the reviewer for their comments and suggestions, which have helped us improve our manuscript.
As pointed out, evaluating the animals’ physiological stability is essential to ensure that cognitive and neuropathological outcomes are not confounded by systemic disruptions. Although we maintained body temperature at 37 °C and confirmed anesthetic depth using reflex-based assessments, we did not perform direct monitoring of parameters such as oxygen saturation, respiratory rate, or arterial blood gases. While no signs of distress or delayed recovery were observed, we acknowledge that the absence of real-time physiological data limits our ability to fully exclude systemic influences.
Following the reviewer’s recommendation, we have addressed this point in the Discussion section as follows:
“Third, we did not perform direct physiological monitoring, such as oxygen saturation, respiratory rate, or arterial blood gas analysis, during the anesthetic period, aside from maintaining body temperature at 37 °C. Although no signs of distress or delayed recovery were observed, and depth of anesthesia was confirmed using righting reflex and withdrawal reflex, the lack of continuous homeostatic data limits our ability to exclude the influence of systemic physiological disruptions on cognitive outcomes [69].”
(page 12, lines 426 – 431)
- The n for behavioral testing is rather low. For negative results it is important to do and show the results of post-hoc power analysis.
We thank the reviewer for their comments and suggestions, which have helped us improve our manuscript.
As pointed out, the sample size for behavioral testing was relatively small, and we agree that a post-hoc power analysis is important, especially when interpreting non-significant results.
Following the reviewer's recommendation, we conducted a post-hoc power analysis using G*Power software (version 3.1.9.4, Heinrich-Heine-Universität Düsseldorf, Germany). Based on an effect size of Cohen’s f = 0.95, α = 0.05, and 8 groups, the analysis indicated that 32 animals in total (4 per group) would be required to achieve 90% power. Since our study included 40 animals (5 per group), this exceeded the minimum requirement. The post-hoc analysis using the same parameters confirmed an actual power of 0.99, indicating that the behavioral analysis was adequately powered to detect meaningful differences.
Following the reviewer’s recommendation, we have revised the text in the Materials and Methods section as follows:
“Power analysis, performed using G*Power software (version 3.1.9.7, Hein-rich-Heine-Universität Düsseldorf, Germany), confirmed that this sample size allowed for the detection of differences between groups under the following conditions: Cohen's f(V) = 0.95, α = 0.05, and 1 - β = 0.90. Specifically, a total of 32 animals (4 per group) would be required to achieve 90% power under these assumptions. A total of 40 mice were used for behavioral testing, with five mice per group sacrificed at each time point. Post hoc power analysis using the same parameters indicated an actual power of 0.99, further confirming that the sample size was adequate for detecting group differences. Following two-way ANOVA, Student’s t-test or Tukey’s HSD post hoc test was performed for comparisons between groups at the same or different time points.”
(page 16, lines 610 – 620)
- It would have been better to have TUNEL staining accompanied by DAPI or some similar cellular marker. These experiments cannot be redone, so this should be acknowledged as a limitation.
We thank the reviewer for their comments and suggestions, which have helped us improve our manuscript.
As pointed out, the use of TUNEL staining without a nuclear counterstain such as DAPI limits the ability to confirm nuclear localization and cellular integrity of apoptotic signals. While this limitation does not invalidate the observed results, it does reduce the resolution and specificity of the apoptosis assessment.
Following the reviewer’s recommendation, we have revised the Discussion section as follows:
“Fourth, although TUNEL staining was used to assess neuronal apoptosis, it was not accompanied by a nuclear counterstain such as 4′,6-diamidino-2-phenylindole (DAPI). The lack of nuclear marker limits the precise localization of DNA fragmentation within cell nuclei and to distinguish apoptotic cells from necrotic or artifactual signals [70]. While the distribution of TUNEL-positive signals supports apoptotic activity, the absence of a nuclear marker reduces the morphological resolution and interpretability of the histological findings. Future histological analyses should include nuclear or cell-specific counterstains to enhance structural validation and accuracy.”
(page 12, lines 432 – 439)
- Can the authors please describe the measures that were taken to ensure that experimenters and those who conducted the analysis were blind to experimental condition.
We thank the reviewer for their comments and suggestions, which have helped us improve our manuscript.
As pointed out, it is essential to describe the blinding procedures used to minimize bias during the experiment and subsequent data analysis.
Following the reviewer's recommendation, we have revised the text in the Methods section as follows:
“To minimize bias, all behavioral testing, histological processing, and data analysis were performed by experimenters who were blinded to the treatment groups. Mice were assigned randomized ID codes independent of group allocation, and group information was withheld from the personnel conducting behavioral assessments and histological quantifications until after data collection was complete.”
(page 13, lines 486 – 491)
- It would be nice to see some reference in discussion to early life anesthesia neurotoxicity and possible relevance of these findings.
We thank the reviewer for their comments and suggestions, which have helped us improve our manuscript.
As pointed out, incorporating a discussion of early-life anesthesia neurotoxicity and its possible relevance to our findings would strengthen the contextual significance of our study.
Following the reviewer’s recommendation, we have revised the text in the Discussion section as follows:
“In addition to findings in older adults or genetically predisposed individuals, early-life exposure to general anesthetics has also been associated with long-term neurodevelopmental impairments in both animal models and human epidemiological studies. Studies in neonatal rodents have demonstrated that exposure to anesthetic agents such as isoflurane, sevoflurane, and propofol during critical periods of brain development can result in widespread neuronal apoptosis, disrupted synaptogenesis, and persistent cognitive deficits [62]. Similarly, epidemiological studies in pediatric populations have reported associations between early exposure to general anesthesia and subsequent impairments in language acquisition, learning ability, and overall cognitive function [63,64]. Although our study primarily focuses on aging and preclinical AD, these findings from early-life models highlight the broader neurotoxic potential of certain anesthetic agents across the lifespan. The shared pathological features suggest that overlapping molecular mechanisms may contribute to anesthesia-induced neurotoxicity in both neurodevelopmental and neurodegenerative contexts.”
(page 11, lines 387 – 400)
- A clear accounting of limitations of the work as the next to last paragraph in the discussion section would improve the overall quality of the wokr.
We thank the reviewer for their comments and suggestions, which have helped us improve our manuscript.
As pointed out, a clear accounting of the study’s limitations has been added as the next-to-last paragraph in the Discussion section. This revised paragraph includes:
(1) a more detailed justification for the use of the MWM and Y-maze tests, which were selected because they are well-validated tools for assessing spatial learning and working memory—cognitive domains frequently impaired in perioperative neurocognitive disorder (PND). We also added suggestions for incorporating additional behavioral paradigms (e.g., novel object recognition) in future studies to better capture the heterogeneity of PND, which can involve deficits in attention, executive function, and recognition memory.
(2) the addition of a limitation regarding the absence of direct physiological monitoring during anesthesia.
(3) the lack of a nuclear counterstain (e.g., DAPI) in TUNEL staining, which limits precise cellular localization.
These revisions strengthen the transparency and critical interpretation of our findings and address the reviewer’s concern regarding the completeness of the study’s limitations.
(page 12, lines 408 – 447)
We have addressed all of the issues raised by the reviewers. We are grateful for the constructive comments made during the review process. We believe that our paper has been improved by implementing these suggestions.
Yours faithfully,
Jong-Hee Sohn, M.D. Ph.D.
Department of Neurology, Chuncheon Sacred Heart Hospital, Hallym University College of Medicine, 77 Sakju-ro, Chuncheon-si, Gangwon-state, 24253, Republic of Korea
Tel: +82-33-252-9970, Fax: +82-33-241-8063
E-mail: deepfoci@hallym.or.kr
Jae-Jun Lee M.D. Ph.D.
Departments of Anesthesiology and Pain medicine, Chuncheon Sacred Heart Hospital, Hallym University College of Medicine, 77 Sakju-ro, Chuncheon-si, Gangwon-state, 24253, Republic of Korea
Tel: +82-33-252-9970, Fax: +82-33-241-8063
E-mail: iloveu59@hallym.or.kr

Reviewer 2 Report
Comments and Suggestions for Authors
The article is entitled " Comparison of cognitive deterioration between propofol and remimazolam anesthesia in ApoE4 knock-in mouse model. " The study's findings are valuable; however, several issues need to be significantly addressed in each section.
Abstract:
- Page 1, line 23; Please specify CA3 and dentate gyrus (DG) regions of the hippocampus in (Neuronal apoptosis and amyloid-beta (Aβ) deposition in the CA3 and dentate gyrus (DG) regions were evaluated preoperatively..)
- In conclusion, it should be added that these findings are associated with more pronounced hippocampal neuropathological changes in Alzheimer's disease
Introduction:
Specify which behavioral tests were used in previous publications (e.g., Morris water maze, Y-maze), as this adds credibility and reproducibility to the methods.
Add paragraph discussingthe mechanism of action and the pharmacological and side effect profile of propofol
Please include a dedicated paragraph elaborating on amyloid-beta (Aβ) deposition, particularly its role in Alzheimer's disease pathogenesis and how it may be influenced by anesthetic agents such as propofol and remimazolam in the ApoE4 mouse model.
Materials and Methods:
- Please specify the name of the university where the study received Institutional Animal Care and Use Committee (IACUC) approval, as referenced in the statement: 'Having received approval from the Institutional Animal Care and Use Committee of our university (No. R1 2021-74).
- Please provide a more clearly organized and detailed description of the experimental timeline, including the specific design of the 8 groups (n = 5 per group) under the two anesthetic conditions—propofol (Group P) and remimazolam (Group R). A visual representation or table of the group assignments, intervention, time of testing, timeline and time of scarifying would also be helpful for clarity
- Page 12, line 43,6 please correct to mice (All surgeries were performed in person by an experienced surgeon, and each surgery lasted approximately 10 minutes).
- In Behavioral Testing: write full name of test in title (4.4.1. MWM Test)
- Page 14, line 514: add full name of (HRP activity)
Results
Page 3, line 114: Move the titles of Figures 1,2, and 3 below the figures.
Conclusion: Add a recommendation
Add study limitations
Author Response
Dear Reviewer 2,
Please find attached a revised version of our manuscript, “Comparison of cognitive deterioration between propofol and remimazolam anesthesia in ApoE4 knock-in mouse model”.
We thank you for your thoughtful suggestions regarding the original version of our paper; most of the suggested changes have been incorporated into the revision.
All revisions are described in detail in the order mentioned in the review, following your comments. We believe that the revisions have greatly improved the manuscript and hereby submit the revised version for publication.
Please note that the revised sections in the manuscript have been highlighted in yellow.
Thank you for your time and consideration.
The article is entitled " Comparison of cognitive deterioration between propofol and remimazolam anesthesia in ApoE4 knock-in mouse model. " The study's findings are valuable; however, several issues need to be significantly addressed in each section.
Abstract:
- Page 1, line 23; Please specify CA3 and dentate gyrus (DG) regions of the hippocampus in (Neuronal apoptosis and amyloid-beta (Aβ) deposition in the CA3 and dentate gyrus (DG) regions were evaluated preoperatively..)
We thank the reviewer for their comments and suggestions, which have helped us improve our manuscript.
As pointed out, specifying the anatomical location of the CA3 and dentate gyrus (DG) regions will improve clarity for readers who may not be familiar with hippocampal subregions.
Following the reviewer’s recommendation, we have revised the sentence in the Abstract as follows:
“neuronal apoptosis and amyloid-beta (Aβ) deposition in the CA3 and dentate gyrus (DG) regions of the hippocampus were evaluated preoperatively and at 2, 4, and 7 days postoperatively.”
(page 1, lines 23 – 24)
- In conclusion, it should be added that these findings are associated with more pronounced hippocampal neuropathological changes in Alzheimer's disease
We thank the reviewer for their comments and suggestions, which have helped us improve our manuscript.
As pointed out, the conclusion of the abstract would benefit from more explicitly linking the findings to Alzheimer's disease–related hippocampal pathology.
Following the reviewer’s recommendation, we have revised the final sentence of the Abstract as follows:
“Conclusions: Propofol was associated with more pronounced neuropathologic changes in the hippocampus compared to remimazolam. These findings suggest that remimazolam may be a safer anesthetic for patients at risk for neurodegenerative disorders, as it is associated with less severe hippocampal pathology, which is characteristic of AD.”
(page 1, lines 30 – 34)
Introduction:
Specify which behavioral tests were used in previous publications (e.g., Morris water maze, Y-maze), as this adds credibility and reproducibility to the methods.
We thank the reviewer for their comments and suggestions, which have helped us improve our manuscript.
As pointed out, specifying which behavioral tests were used in previous publications improves the clarity and reproducibility of our methods and provides stronger support for our experimental design.
Following the reviewer’s recommendation, we have revised the text in the Introduction as follows:
“Previous results have shown that propofol induces persistent cognitive deficits and hippocampal pathology in pre-symptomatic AD mice [21]. These effects were validated through behavioral tests, notably the Morris Water Maze (MWM) and Y-Maze, which are widely used in preclinical models to assess hippocampal function. The MWM evaluates spatial memory dependent on hippocampal integrity [29], while the Y-Maze measures short-term memory and exploratory behavior [30]. Given that these cognitive domains are frequently impaired in PND [31], both tasks are commonly employed to investigate anesthesia-related cognitive dysfunction.”
(page 2, lines 81 – 88)
Add paragraph discussingthe mechanism of action and the pharmacological and side effect profile of propofol
We thank the reviewer for their comments and suggestions, which have helped us improve our manuscript.
As pointed out, adding a paragraph describing the mechanism of action and pharmacological and side effect profile of propofol would improve the background and scientific rationale of the study.
Following the reviewer's recommendation, we have revised the text in the Introduction as follows:
“Propofol is a widely used intravenous anesthetic agent known for its rapid onset and short duration of action. It primarily acts as a positive allosteric modulator of the gamma-aminobutyric acid type A (GABAA) receptor, enhancing inhibitory neurotransmission and inducing sedation, hypnosis, and amnesia [23]. Despite its favorable pharmacokinetics, propofol has been associated with several adverse effects, including hypotension, respiratory depression, and dose-dependent neurotoxicity [24,25]. Preclinical studies have shown that propofol may exacerbate Aβ aggregation, induce tau hyperphosphorylation, and trigger neuronal apoptosis, particularly in models predisposed to AD pathology [21,26-28].”
(page 2, lines 72 – 80)
Please include a dedicated paragraph elaborating on amyloid-beta (Aβ) deposition, particularly its role in Alzheimer's disease pathogenesis and how it may be influenced by anesthetic agents such as propofol and remimazolam in the ApoE4 mouse model.
We thank the reviewer for their comments and suggestions, which have helped us improve our manuscript.
As pointed out, we have added a paragraph detailing Aβ deposition, its role in Alzheimer’s disease pathogenesis, and how anesthetic agents may modulate this process.
Propofol has been shown to exacerbate Aβ aggregation and AD-related pathology in preclinical models, while some studies suggest remimazolam may also increase Aβ levels via glutamate excitotoxicity.
However, no studies have investigated the effects of remimazolam in the ApoE4 mouse model, and direct comparisons between remimazolam and propofol regarding Aβ pathology are lacking. This addition strengthens the mechanistic rationale for our study.
Following the reviewer's recommendation, we have revised the text in the Introduction as follows:
“ApoE4 is thought to influence amyloid-beta (Aβ) aggregation and clearance, tau phosphorylation, and neuroinflammation, thereby modulating the vulnerability of the brain to neurodegenerative processes [16]. Aβ, a cleavage product of the amyloid precursor protein (APP), accumulates abnormally in the brain in AD and is associated with synaptic dysfunction and neuronal toxicity [17]. Aβ peptides aggregate to form oligomers and plaques that disrupt synaptic function, induce oxidative stress, and trigger neuroinflammatory responses, ultimately leading to neuronal death and cognitive impairment [18].”
(page 2, lines 57 – 64)
“Preclinical studies have shown that propofol may exacerbate Aβ aggregation, induce tau hyperphosphorylation, and trigger neuronal apoptosis, particularly in models predisposed to AD pathology [21,26-28].”
(page 2, lines 77 – 80)
“However, recent studies suggest it may induce cognitive dysfunction via glutamate excitotoxicity and increase Aβ levels [35]. Despite these findings, no study has directly compared the extent of Aβ accumulation between remimazolam and propofol, leaving their relative neurotoxicity unclear.”
(page 3, lines 96 – 100)
Materials and Methods:
- Please specify the name of the university where the study received Institutional Animal Care and Use Committee (IACUC) approval, as referenced in the statement: 'Having received approval from the Institutional Animal Care and Use Committee of our university (No. R1 2021-74).
We thank the reviewer for their comments and suggestions, which have helped us improve our manuscript.
As pointed out, specifying the name of the institution that granted IACUC approval improves transparency and ethical accountability.
Following the reviewer’s recommendation, we have revised the text in the Methods section as follows:
“Having received approval from the Institutional Animal Care and Use Committee of Hallym university (No. R1 2021-74), all experimental procedures in this study were performed in accordance with the committee's guidelines.”
(page 13, lines 456 – 457)
- Please provide a more clearly organized and detailed description of the experimental timeline, including the specific design of the 8 groups (n = 5 per group) under the two anesthetic conditions—propofol (Group P) and remimazolam (Group R). A visual representation or table of the group assignments, intervention, time of testing, timeline and time of scarifying would also be helpful for clarity
We thank the reviewer for their comments and suggestions, which have helped us improve our manuscript.
As pointed out, the experimental timeline and group allocation required a clearer and more detailed explanation, including the specific design of the eight groups under the two anesthetic conditions.
Following the reviewer’s recommendation, we have revised the text in the Methods as follows:
“The overall experimental timeline and group design are shown in Figure 4.”
(page 13, lines 485 – 486)
Following the reviewer’s recommendation, we have added Figure4 and Figure legend as follows:
“Figure 4. Experimental timeline and group allocation. All mice underwent 5 days of pre-training with twice-daily Morris water-maze (MWM) sessions. On the preoperative day, Y-maze and MWM tests were conducted, after which mice were sacrificed (n = 5 per group) for histological analysis. The remaining mice underwent surgery, under either propofol (Group P) or remimazolam (Group R) anesthesia. Postoperatively, behavioral tests and sacrifice were conducted on days 2, 4, and 7 (n = 5 per group at each time point).”
(page 13, lines 492 – 498)
- Page 12, line 43,6 please correct to mice (All surgeries were performed in person by an experienced surgeon, and each surgery lasted approximately 10 minutes).
We thank the reviewer for their comments and suggestions, which have helped us improve our manuscript.
As pointed out, the sentence structure should be revised to clarify that surgeries were performed on mice, not by them.
Following the reviewer’s recommendation, we have revised the text in the Methods section as follows:
“All surgeries on mice were performed by an experienced surgeon, and each surgery lasted approximately 10 minutes.”
(page 14, lines 521 – 523)
- In Behavioral Testing: write full name of test in title (4.4.1. MWM Test)
We thank the reviewer for their comments and suggestions, which have helped us improve our manuscript.
As pointed out, providing the full name of the test in the section title improves clarity for readers who may not be familiar with abbreviations.
Following the reviewer’s recommendation, we have revised the text in the Methods section as follows:
“4.4.1. Morris Water Maze Test”
(page 14, lines 536)
- Page 14, line 514: add full name of (HRP activity)
We thank the reviewer for their comments and suggestions, which have helped us improve our manuscript.
As pointed out, adding the full name of "HRP" improves clarity for readers unfamiliar with the abbreviation.
Following the reviewer’s recommendation, we have revised the text in the Methods section as follows:
“Sections were subsequently incubated with a horseradish peroxidase (HRP)-conjugated secondary antibody (1:250 dilution, Thermo Fisher Scientific Inc., Waltham, MA, USA) for 1 h and rewashed. To visualize HRP activity, 3,3′-Diaminobenzidine (DAB) was used as the chromogen.”
(page 16, lines 598 – 601)
Results
Page 3, line 114: Move the titles of Figures 1,2, and 3 below the figures.
We thank the reviewer for their comments and suggestions, which have helped us improve our manuscript.
As pointed out, placing figure titles below the figures enhances readability and formatting consistency.
Following the reviewer’s recommendation, we have revised the layout of the manuscript so that:
The titles of Figures 1, 2, 3, and 4 have been moved to appear directly below each corresponding figure.
Conclusion: Add a recommendation
We thank the reviewer for their comments and suggestions, which have helped us improve our manuscript.
As pointed out, a clear recommendation enhances the applicability of our findings. A recommendation has therefore been added to the Conclusion section to emphasize the potential clinical implications of our results.
Following the reviewer’s recommendation, we have revised the text in the Conclusion as follows:
“In conclusion, while both remimazolam and propofol contribute to postoperative neurocognitive decline, the severity of neurodegenerative changes appears to be greater with propofol. The differences in histopathologic changes suggest that remimazolam may be a safer anesthetic option for patients with cognitive impairment. Based on these findings, it is recommended that anesthetic choice be carefully considered in patients at risk for neurodegeneration. Further clinical studies are warranted to confirm these results and explore long-term cognitive outcomes in at-risk human populations.”
(page 16, lines 628 – 631)
Add study limitations
We thank the reviewer for their comments and suggestions, which have helped us improve our manuscript.
As pointed out, a clear accounting of the study’s limitations has been added to the Discussion section to enhance the transparency and interpretability of our findings. This revised paragraph includes:
(1) a more detailed justification for the use of the MWM and Y-maze tests, which were selected because they are well-validated tools for assessing spatial learning and working memory—cognitive domains frequently impaired in perioperative neurocognitive disorder (PND). We also added suggestions for incorporating additional behavioral paradigms (e.g., novel object recognition) in future studies to better capture the heterogeneity of PND, which can involve deficits in attention, executive function, and recognition memory.
(2) the addition of a limitation regarding the absence of direct physiological monitoring during anesthesia.
(3) the lack of a nuclear counterstain (e.g., DAPI) in TUNEL staining, which limits precise cellular localization.
These revisions strengthen the transparency and critical interpretation of our findings and address the reviewer’s concern regarding the completeness of the study’s limitations.
“Our study has several limitations. First, there is the relatively short observation period, focusing on cognitive function and neuropathologic changes up to 7 days after surgery. Although significant differences between remimazolam and propofol were observed during this period, it remains uncertain whether these cognitive effects persist or evolve over a longer period. Given that PND can manifest weeks to months after surgery, this short-term assessment may not fully capture the long-term cognitive impact of these anesthetics. Future research should extend the observation period to assess whether the initial cognitive impairments resolve, persist, or worsen over time. Second, the study relied on a limited number of behavioral assessments, specifically the MWM and Y-maze tests, which access hippocampal-dependent spatial and short-term memory, respectively [44]. These paradigms were selected because they are well-established, sensitive, and commonly used to assess cognitive domains frequently affected in PND. However, PND is a heterogeneous condition that also involves other domains such as attention, executive function, and learning. The inclusion of additional behavioral tests, such as the novel object recognition test [45], which assesses recognition memory and is less reliant on spatial navigation, could have helped identify non-hippocampal cognitive deficits. Expanding the cognitive testing battery in future studies would allow for a deeper exploration of the cognitive changes associated with anesthesia and surgery. Third, we did not perform direct physiological monitoring, such as oxygen saturation, respiratory rate, or arterial blood gas analysis, during the anesthetic period, aside from maintaining body temperature at 37 °C. Although no signs of distress or delayed recovery were observed, and depth of anesthesia was confirmed using righting reflex and withdrawal reflex, the lack of continuous homeostatic data limits our ability to exclude the influence of systemic physiological disruptions on cognitive outcomes [ref]. Fourth, although TUNEL staining was used to assess neuronal apoptosis, it was not accompanied by a nuclear counterstain such as 4′,6-diamidino-2-phenylindole (DAPI) [ref]. The lack of nuclear marker limits the precise localization of DNA fragmentation within cell nuclei and to distinguish apoptotic cells from necrotic or artifactual signals. While the distribution of TUNEL-positive signals supports apoptotic activity, the absence of a nuclear marker reduces the morphological resolution and interpretability of the histological findings. Future histological analyses should include nuclear or cell-specific counterstains to enhance structural validation and accuracy. Fifth, this study primarily focuses on key neuropathological markers like Aβ deposition and neuronal apoptosis but does not address other important molecular mechanisms, such as neuroinflammation and oxidative stress, which are known to play significant roles in cognitive decline after surgery [46]. The absence of data on neuroinflammatory cytokines or markers of oxidative stress limits the ability to draw firm conclusions about the underlying mechanisms driving the observed cognitive impairments. Future studies should integrate these molecular analyses to better understand the broader biological processes contributing to anesthesia-induced cognitive decline.”
(page 12, lines 408 – 667)
We have addressed all of the issues raised by the reviewers. We are grateful for the constructive comments made during the review process. We believe that our paper has been improved by implementing these suggestions.
Yours faithfully,
Jong-Hee Sohn, M.D. Ph.D.
Department of Neurology, Chuncheon Sacred Heart Hospital, Hallym University College of Medicine, 77 Sakju-ro, Chuncheon-si, Gangwon-state, 24253, Republic of Korea
Tel: +82-33-252-9970, Fax: +82-33-241-8063
E-mail: deepfoci@hallym.or.kr
Jae-Jun Lee M.D. Ph.D.
Departments of Anesthesiology and Pain medicine, Chuncheon Sacred Heart Hospital, Hallym University College of Medicine, 77 Sakju-ro, Chuncheon-si, Gangwon-state, 24253, Republic of Korea
Tel: +82-33-252-9970, Fax: +82-33-241-8063
E-mail: iloveu59@hallym.or.kr

Round 2
Reviewer 2 Report
Comments and Suggestions for Authors
Many thanks for authors all inquiries were addressed